# The Genetics and Breeding of Heat Stress Tolerance in Wheat: Advances and Prospects

**DOI:** 10.3390/plants14020148

**Published:** 2025-01-07

**Authors:** Yuling Zheng, Zhenyu Cai, Zheng Wang, Tagarika Munyaradzi Maruza, Guoping Zhang

**Affiliations:** Key Laboratory of Crop Germplasm Resource of Zhejiang Province, Department of Agronomy, Zhejiang University, Hangzhou 310058, China; 22316030@zju.edu.cn (Y.Z.); 3200101462@zju.edu.cn (Z.C.); 12316085@zju.edu.cn (Z.W.); tag.maruza@gmail.com (T.M.M.)

**Keywords:** climate change, heat stress, wheat, genetic basis, thermo-tolerant gene, breeding

## Abstract

Heat stress is one of the major concerns for wheat production worldwide. Morphological parameters such as germination, leaf area, shoot, and root growth are affected by heat stress, with affected physiological parameters including photosynthesis, respiration, and water relation. Heat stress also leads to the generation of reactive oxygen species that disrupt the membrane systems of thylakoids, chloroplasts, and the plasma membrane. The deactivation of the photosystems, reduction in photosynthesis, and inactivation of Rubisco affect the production of photo-assimilates and their allocation, consequently resulting in reduced grain yield and quality. The development of thermo-tolerant wheat varieties is the most efficient and fundamental approach for coping with global warming. This review provides a comprehensive overview of various aspects related to heat stress tolerance in wheat, including damages caused by heat stress, mechanisms of heat stress tolerance, genes or QTLs regulating heat stress tolerance, and the methodologies of breeding wheat cultivars with high heat stress tolerance. Such insights are essential for developing thermo-tolerant wheat cultivars with high yield potential in response to an increasingly warmer environment.

## 1. Introduction

Global warming has been a climatic issue for several years and has currently intensified. It was reported that the average global temperature increased by 1.04 °C between 1880 and 2019 [1] and will increase by 2–4 °C by the end of this century [2]. The change in global temperature has a resounding impact on crop production. When the global average temperature rises by 1 °C, the grain yield of wheat (*Triticum aestivum* L.) decreases by 6.0%, rice (*Oryza sativa* L.) by 3.2%, corn (*Zea mays* L.) by 7.4%, and soybean (*Glycine max* Merr.) by 3.1% on average [3]. Heat stress (HS) caused by high temperatures has emerged as a major threat to crop production worldwide [4].

Wheat is one of the most widely grown crops in the world and contributes about 20% of the total dietary calories and proteins worldwide, playing a vital role in food security [5]. The world population is expected to reach approximately 10 billion by 2050, and demand for food will increase by 77.0% over the same period [6,7]. Nevertheless, exceptionally high temperatures during the growth of wheat will reduce the yield potential of wheat in numerous regions of the world [8].

The increasingly high temperature caused by global climate change has brought great challenges to sustainable wheat production as well as food security in the world. It is imperative to develop high-yielding and climate-resilient wheat varieties that can cope with challenging environmental conditions, ensuring food security for the growing population.

## 2. Impacts of Heat Stress on Wheat Growth and Yield Formation

Heat stress affects growth and development processes in wheat [9], resulting in an alteration of growth and development patterns, changes in physiological functions, and a reduction in grain yield. High temperatures have a direct influence on the photosynthetic apparatus of wheat leaves, leading to a decline in photosynthesis and biomass production [10] while also shortening the vegetative period, reducing tillering capacity and spikelet differentiation. In addition, heat stress induces oxidative stress, inhibiting growth and promoting senescence [11].

### 2.1. Morphology and Growth

High temperature affects vegetative growth and biomass production, resulting in the alteration of organ or tissue development and genesis. High temperatures can result in a 44% reduction in the total biomass of wheat during the reproduction stage [12]. Heat stress occurring at the vegetative stage shortens the duration of vegetative growth and reduces leaf area and tillers per plant. If heat stress is experienced at the generative stage, it causes leaf senescence and a reduction in grains per spike and kernel weight [13]. Heat stress conditions where seedlings were exposed to 45 °C for 2 h after 7 days of their germination displayed significantly reduced shoot and root dry mass, shoot length, and root length. Additionally, there was a decrease in the chlorophyll content and membrane stability index, while the proline content and antioxidants significantly increased [14].

Leaf senescence is one of the inimitable symptoms when wheat plants are exposed to high temperature, characterized by structural changes of chloroplasts, followed by a vacuolar collapse, and ultimately a loss of plasma membrane integrity and interference with cellular homeostasis [15]. In many studies, it has been shown that leaf yellowing or leaf chlorosis is one of the earliest symptoms of premature leaf senescence, which is caused by heat-induced chlorophyll degradation or heat-inhibited chlorophyll biosynthesis [16]. When leaf chlorophyll content reduces, leaf senescence accelerates as a result of the impact of heat stress [17,18]. Under heat stress (>34 °C), chlorophyll biosynthesis in wheat is greatly inhibited, and senescence is enhanced [19].

### 2.2. Physiological and Biochemical Activities

#### 2.2.1. Water Relations

Wheat crops exposed to high temperatures (35/25 °C) after tillering experience a significant reduction in water potential, with a greater decrease observed in genotypes susceptible to heat stress compared to heat-tolerant genotypes [20]. Heat stress also increases the hydraulic conductivity of cell membranes and plant tissues, primarily attributed to increased aquaporin activity [21] and, to a greater extent, reduced water viscosity [22]. With a concomitant increase in leaf temperature, wheat plants exposed to heat stress substantially decrease the water potential and the relative water content in leaves, leading to reduced photosynthetic productivity [23].

#### 2.2.2. Photosynthesis

Photosynthesis is the most sensitive physiological parameter affected by heat stress, leading to poor growth performance in wheat [24]. A major effect of heat stress is the reduction in photosynthesis resulting from decreased leaf area expansion, impaired photosynthetic machinery, and premature leaf senescence ultimately leading to reduced wheat production [25,26]. In the grain-filling stage, a day–night temperature swing of 34/26 °C for 16 days decreased grain yield by 43% due to a decrease in individual grain weight of 44%, associated with a decrease in Fv/Fm [27]. Heat stress causes disruption of thylakoid membranes, thereby inhibiting the activities of membrane-associated electron carriers and enzymes, ultimately reducing the photosynthetic rate [28]. The impediment of photosynthetic activities may also be attributed to the reduced soluble protein Rubisco and its binding proteins [29,30]. The key regulatory enzyme of Rubisco, Rubisco activase, is reported to be dissociated above 30 °C, causing a reduction in the photosynthetic capacity of wheat leaves [31]. In photosynthesizing tissues, photosystem-II is more sensitive to heat stress than photosystem-I.

#### 2.2.3. Respiration

The temperature coefficient of respiration, Q10, describes a proportional increase in respiration rate with every 10 °C rise in temperature [32]. The respiration rate increases with increasing temperature, but at a certain level of temperature, it diminishes due to damage to the respiratory apparatus [33]. Research has shown that the respiration rate in wheat plants rapidly increases at 30 °C to 35 °C, while the photosynthetic rate rapidly declines [34], leading to a marked reduction in biomass production. The differential influence of heat stress on photosynthesis and respiration is attributed to the different organelles and enzyme systems associated with each respective process [9]. Moreover, it was found that heat-susceptible varieties had higher respiration rates compared to tolerant varieties under high temperatures [35]. HS also increases photorespiration and decreases membrane stability [36]. Photorespiration is enhanced by the presence of high oxygen concentrations [37], and the process can dissolve excess ROS, consume ATPs and NADPH, and reduce glyoxylate generated photosynthetically [38,39]. These pathways can cumulatively lead to a 20% yield reduction at most in wheat [40].

### 2.3. Grain Yield and Quality

#### 2.3.1. Grain Yield

At high temperatures, wheat grains fill more quickly but for a shorter period of time. However, quick grain-filling rates have not been able to compensate for the shorter time for assimilate accumulation when subjected to heat stress [41], leading to reduced kernel weight. Moreover, the reduced grain filling at high temperatures is also attributed to fewer assimilates and less remobilization of stem reserves.

Day and night temperatures of 37 °C and 28 °C, respectively, lasting for about 10 to 20 days, resulted in a yield reduction characterized by a shortening of the filling and maturation times of grains, diminished fresh and dry weight, and a reduction in protein and starch contents [37]. For a temperature increase of 2 °C, rising temperatures are projected to have a variety of effects on falling wheat yields ranging from 1% to 28%. For a temperature increase of 4 °C, the range extends from 6% to 55% [42]. When temperatures rise above 31 °C just before anthesis, pollen sterility occurs and decreases the seed-setting rate, subsequently affecting yield and yield components [40]. Wheat is less likely to bounce back if harmed during the flowering stage as this stage is most susceptible to high temperatures [43]. This vulnerability is of chief concern given that grains per spike is most significantly correlated with grain yield [44]. Additionally, heat stress during grain filling results in enhanced leaf senescence rates and reduced grain-filling duration [45,46].

#### 2.3.2. Grain Quality

Under HS, starch content in wheat endosperm is greatly reduced [47]. A drop in starch content, which accounts for more than 65% of grain dry weight, consequently results in yield reduction [48]. Meanwhile, HS during grain filling can have a detrimental influence on protein content in grains as it reduces starch deposition [49]. HS disrupts the balance of nitrogen and starch in wheat grains, allocating relatively more nitrogen toward the formation of protein, leading to an increase in protein concentration [50]. Exposure to heat stress also reduces glutenin synthesis, but gliadin synthesis remains unchanged or increases [51]. Heat stress can also lead to a reduction in essential amino acids along with an increase in protein levels, which can impact the sedimentation index, a measure of the grain’s protein quality [52]. Flour produced from grain grown under heat stress tends to have reduced consistency due to decreased gluten strength-related parameters, such as lactic acid retention ability and mixograph peak time [53]. Moreover, HS also decreases the swelling strength of wheat flour noodles and increases the amount of broken grains [54].

## 3. The Responses of Wheat to Heat Stress

Sessile plants evolved defense systems to deal with environmental challenges, including immediate avoidance and long-term tolerance. These defense systems provide plants with heat stress tolerance by preserving and repairing damaged proteins and membranes [36]. This evolutionary capability allows for the production of economic yield at a temperature above the threshold [55].

### 3.1. Antioxidant Defense System

Exposure of plants to heat stress often leads to the generation of destructive reactive oxygen species (ROS), responsible for generating oxidative stress, which in turn promote protein denaturation and unsaturated fatty acid production, ultimately increasing membrane peroxidation and decreasing membrane thermo-stability [43]. However, ROS may also act as a signaling molecule under unfavorable abiotic conditions, promoting resistance to adverse conditions.

The antioxidant defense mechanism is accountable for maintaining the balance of ROS production and detoxification in plants. There are mainly two types of antioxidant defense systems found in wheat, i.e., enzymatic and non-enzymatic [56]. Photorespiration can degrade excess ROS both directly and indirectly [38]. The conversion of ROS to O_2_ and water depends on the superoxide dismutase (SOD), catalase (CAT), and guaiacol peroxidase (POX) systems. Antioxidant enzyme levels increased significantly when wheat seedlings were exposed to short-term heat stress (45 °C for 2 h) and are highly correlated with other heat tolerance traits [14]. Heat-stress-tolerant wheat varieties demonstrate enhanced glutathione-S-transferase (GST), ascorbate peroxidase (APX), and CAT activities and protection against heat stress injuries [57]. Furthermore, superoxide radicals reduce metal ions in cells through spontaneous dismutation or catalytic activity of SOD [58].

### 3.2. Photosystems and Chlorophyll Content

Besides antioxidant defense systems, plants utilize other mechanisms to protect the photosystems such as cyclic electron flow (CEF), the alternative oxidase (AOX) pathway, oxidative electron transport, and mitochondrial reactions of photorespiration [59]. The photorespiration reactions serve as direct sinks for ATP, NADPH, and reduced ferredoxin generated photosynthetically [39]. On the other hand, the peroxisomal catalase scavenging of H_2_O_2_, CEF optimization, promotion of the AOX pathway, and CO_2_ release from glycine decarboxylation for intracellular recycling are the indirect ways in which the photosystems are protected [60]. Additionally, photorespiratory metabolism generates glycine as a source of glutathione, which acts as a major antioxidant in plant cells [61].

High chlorophyll content, associated with transpiration efficiency under heat stress, conveys a low degree of photoinhibition. As a result, it is considered a desirable trait for heat tolerance in wheat [62]. A significantly positive relationship has been found between leaf chlorophyll content and transpiration efficiency in heat-tolerant genotypes [63].

Chloroplasts play a significant role in the activation of signaling pathways in response to cellular stimuli under heat stress, aiding in the induction of the expression of nuclear heat-response genes [64]. Heat stress response in the nucleus requires the translation of chloroplast protein to stimulate retrograde signaling [65]. Retrograde signaling can be defined as the communication pathway wherein nuclear transcriptional activities are regulated in part by signals derived from plastids and/or mitochondria. Retrograde signaling largely includes developmental control of organelle biogenesis and operational control to acclimate to environmental stresses [66]. Chloroplasts act as a specialized sensor of intra- and extracellular stimuli and combine a variety of intracellular signals and pathways for sustaining homeostasis both at cellular and organismal levels [67].

### 3.3. Canopy Temperature Depression and Mobilization of Stem Reserves

Under heat stress conditions, around 7% to 9% grain yield can be achieved through canopy temperature depression and remobilization of stem carbohydrates [68]. The term canopy temperature depression (CTD) describes the deviation of plant/crop canopy temperature from ambient temperature [69]. CTD acts as a good indicator of a genotype’s fitness under heat stress, playing an important role in sustaining the physiological basis of grain yield when wheat is exposed to heat stress [70].

An effective heat tolerance mechanism in wheat is the enhanced mobilization of stem reserves [71]. Around 90% to 95% of the carbon needed for grain filling is achieved from current carbon assimilation under ideal conditions [72]. However, heat stress reduces the translocation of assimilates from the photosynthetic sources, prompting the remobilization of alternative sources such as stem reserves for grain filling [73]. Stem reserve mobilization contributes 75–100% to grain yield under HS [74], as this mobilization is strongly associated with carbohydrate metabolism [75]. Early maturing wheat genotypes having an efficient remobilization capacity of the stem carbohydrate reserves can be considered valuable [76], also exhibiting rapid ground cover and accelerated grain-filling responses to minimize the severe effects of terminal heat stress [77].

### 3.4. Hormone-Mediated Regulation

In plants, stress resilience is conferred by a complex network of physiological, biochemical, sub-atomic, and hormonal systems [78], all of which help reduce the harmful impact of HS on crop growth and development [79].

It is well established that abscisic acid (ABA), which governs stomata closure and water intake, improves water use efficiency and drought tolerance [80]. ABA is essential for stomata closure, preventing excessive water loss under dry and salt stress [81]. It also activates signaling pathways and activates regulatory genes that enable the plant to adapt to abiotic challenges such as heat stress [82].

Auxin and cytokinins regulate source photosynthate/nutrient remobilization, which is essential for cereal grain filling and development [83]. Auxin upregulation improves sink capacity and nutrient assimilation. Plant-produced cytokinins, hormones that influence cell division and growth, provide defense against high temperature [84]. They achieve this by boosting photosynthesis, delaying leaf senescence, and improving water use efficiency [85]. Additionally, they also regulate heat shock proteins, which protect plants from heat damage [86].

Another plant hormone, ethylene, stimulates crop development by boosting heat stress sensitivity gene expression and influencing fruit ripening [87,88]. It has been experimentally demonstrated that gibberellins can enhance crop development under high temperature, but their effectiveness depends on the crop species and heat stress severity [89]. Hormones like salicylic acid increase heat tolerance and decrease oxidative harm by promoting the movement of cell reinforcement chemicals [90,91].

### 3.5. Heat Shock Response

The heat shock response (HSR) is a natural mechanism through which plant tissues respond to HS by momentary gene expression reprogramming patterns [92]. Two essential components, the timely perception of stress and the signal transduction cascade, are necessary for a plant to respond well to a stress tolerance mechanism and survive [93]. In plant cells, the plasma membrane acts as the primary sensor, enabling early detection of small temperature changes and stimulating the temporary opening and depolarization of certain heat-sensitive Ca^2+^ channels [94]. Numerous signaling pathways and their components have been found through two-way genomic analysis and gene expression research [95]. The cell redox system plays a significant role in stress signaling, and genome reprogramming triggers biological signaling pathways that include ROS, Ca^2+^, and hormone production by plants [96]. Temperature change causes a physical state transition in the membrane, which is crucial for detecting and controlling gene expression. The expression patterns of numerous enzymes are ultimately impacted by the multiple membrane-level changes caused by HS, including thylakoid membrane rigidification and a change in the ratio of saturated to unsaturated fats [97,98]. Under extreme temperatures, Ca^2+^ ions are essential for temperature sensing and signaling (Figure 1).

## 4. Molecular Mechanism of Heat Stress Tolerance in Wheat

In order to cope with HS, plants implement various regulatory mechanisms at the molecular level. During stress, the plant’s response system, consisting of transcription factors (TFs) and heat shock proteins (HSPs), helps scavenge accumulated ROS, thereby sustaining metabolic activities and production.

### 4.1. Heat Shock Proteins

Heat stress produces stressors that disrupt critical metabolic processes such as DNA replication, transcription, protein transport, and translation. HSPs play a crucial role under heat stress, binding to denatured proteins, preventing protein aggregation and facilitating their reformation under favorable temperatures [99]. As molecular chaperones, these proteins stabilize partially unfolded or denatured proteins and prevent protein denaturation and aggregation during HS [100]. HSPs have other various functions related to heat stress, including acting as transcriptional activators and regulating gene expression through mechanisms like temperature sensing, signal transfer, and binding to DNA [101].

HSP20, HSP60, HSP70, HSP90, and HSP100 are five HSPs with distinct characteristics [102]. The upregulation of HSP70s and cytoskeletal proteins in pollen tissues is linked with fertility restoration under hot environments [103]. *HSP70* expression is also positively correlated with total antioxidant capacity and negatively correlated with cell membrane stability. While HSP60 and HSP70 are highly conserved specialized proteins dedicated to combating HS, HSP20 directs the destruction of improperly folded proteins [104]. HSP90, sometimes referred to as ClpB, is involved in the trafficking and activation of signaling proteins during HS. Under high-temperature conditions (37 °C and 42 °C), the *TaHsp90* gene is expressed at a 7.6 times higher level in the heat- and drought-tolerant Indian wheat cultivar C−306 [105]. HSP100 aids in correct protein folding and disaggregation [106]. High levels of HSP100 are found in developing grains of heat-tolerant wheat cultivars compared to heat-susceptible cultivars under HS treatment [107]. A Single-Nucleotide Polymorphism (SNP) in the heat shock protein HSP16.9 contributes to a 29.89% phenotypic difference in grain weight per spike between heat-resistant and heat-susceptible wheat genotypes [108]. Most HSPs synthesized by eukaryote organisms have six different structures, namely HSP100, HSP90, HSP70, HSP60, HSP40, and Small HSP (SmHSP) found in the nucleus, mitochondria, chloroplast, endoplasmic reticulum, and cytosol (Table 1). There are 753 *HSP* genes known to exist in the wheat genome, including 169 *TaSHSPs*, 273 *TaHSP40s*, 95 *TaHSP60s*, 114 *TaHSP70s*, 18 *TaHSP90s*, and 84 *TaHSP100s* [102].

### 4.2. Transcription Factors Associated with Heat Shock Response

Heat shock transcription factors (HSFs) are the central regulators of HSP expression and are the principal regulators of HSR [121]. Under normal conditions, HSFs exist in a monomeric state, and their activities are repressed by inhibitory association with HSPs, such as HSP70 [122]. However, in the event of HS, the HSPs are detached from HSFs and bind to misfolded/unfolded proteins. The released HSFs then trimerize, undergo phosphorylation, and enter the nucleus. The trimerized HSFs then bind to heat shock elements (HSEs), present in the promoters of target genes, to activate HSR [123]. Plants contain multiple *HSF* members in their genome, with wheat harboring 61 *HSF* genes [124]. Plant HSFs are classified into three classes (A, B, and C) based on the distinctive features of their hydrophobic associated A/B (HR-A/B) region. Both class A and C HSFs contain amino acid residue insertions between their A and B regions. Class A contains 21 residues, while class C contains 7 [125]. The overexpression of heat shock transcription factor *TaHsfA6f* in wheat upregulates multiple *HSPs* and other heat stress defense proteins such as Golgi anti-apoptotic protein (GAAP) and the broad Rubisco-activase isoform [126].

### 4.3. The Roles of Epigenetics

Genome-wide analysis of DNA methylation in wheat revealed that heat stress has a small but striking effect on gene expression. However, in some cases, methylation is associated with small changes in the expression of important genes during heat stress [127], indicating that DNA methylation is associated with alterations in heat-stress-responsive genes and deserves further exploration. So far, 52 wheat cytosine-5 DNA methyltransferases (C5-MTases) have been identified, most of them responsive to both drought stress and heat stress [128].

Moreover, non-coding RNAs have been reported to participate in regulating heat response in wheat [129]. For example, *TamiR159* is downregulated when a heat-sensitive wheat genotype is exposed to a 2 h heat treatment. *TamiR159* targets *TaGAMYB1* and *TaGAMYB2* and directs their cleavage. Overexpression of TamiR159 in rice causes increased heat sensitivity compared with wild type [130]. In addition, 77 differentially expressed long non-coding RNAs were identified before and after heat stress, parts of which are speculated to function as siRNAs [131].

MicroRNAs (miRNAs) are non-coding small RNA that serve as the regulation of post-transcriptional gene expression in plants. The role of miRNAs in the heat-stress-related signaling pathway has also been reported in wheat [132]. Recently, degraded sequence analysis of small RNAs identified and validated heat-stress-regulated miRNAs and their target genes in wheat [132]. In total, 202 miRNAs with 36 miRNAs differentially expressed upon heat stress were identified. Furthermore, observation revealed some of the miRNA targets included heat stress response genes. For instance, miR156 targets SPLs protein, miR159 targets MYB transcription factor, and miR398 regulates superoxide dismutase [132].

## 5. Breeding for Heat Tolerance

The discovery of heat-stress-mediated morphological, physiological, and molecular responses has guided exhaustive research on how plants combat heat stress by inherent genetic variation or creating artificial variations using genome editing or mutational breeding [133]. Researchers emphasize the need to integrate heat stress recovery into breeding programs to complement recent progress in improving plant heat stress tolerance [134].

### 5.1. Genes or QTLs Regulating Heat Stress Tolerance

Heat tolerance is a quantitative trait that is governed by many minor quantitative trait loci (QTLs) [135]. Genetics associated with heat stress have been studied in the past, with many studies trying to map genetic loci controlling heat stress tolerance in wheat. Using Langdon chromosome substitution lines (CSLs), the first mapping investigation for heat tolerance was carried out in the 1990s. The genes responsible for heat tolerance were located on chromosomes 3A, 3B, 4A, 4B, and 6A [136]. Chromosomes 3A, 3B, and 3D were later found to correlate with heat tolerance in the wheat cultivar Hope [137]. Three heat tolerance QTLs were detected on chromosomes 1B, 5B, and 7B in relation to the heat susceptibility index (HSI) by examining 144 recombinant inbred lines (RILs) with varied heat sensitivities derived from cultivars Kauz and MTRWA116 [138]. The HSI of the yield component of an RIL population of wheat was analyzed with the heat-tolerant parent Halberd and heat-sensitive parent Cutter under controlled HS environments (38 °C day/18 °C night). Twenty-seven QTLs associated with improved heat tolerance were detected, and five (located on chromosomes 1A, 2A, 2B, and 3B) were consistently detected in two-year experiments [139]. Developments in quantitative genetics and molecular markers offer potential tools for identifying QTLs influencing heat tolerance in wheat [140]. Eight major QTL regions showing association with drought and heat tolerance located on chromosomes 1B, 2B, 2D, 4A, 4B, 4D, 5A, and 7A were recorded in a meta-QTL study [141]. Sangwan et al. developed a recombinant inbred line (RIL) population derived from the WH1021 (heat-tolerant) and WH711 (heat-sensitive) varieties. Significant genomic regions associated with heat tolerance were detected on chromosomes 2A, 2D, 4A, and 5A with a consistent QTL found on chromosome 2D based on photosynthetic rate analysis [142]. With the availability of the published sequencing data, progress in map-based cloning for heat tolerance can be achieved for cloning major QTLs [143]. Genome-wide association analysis (GWAS) has been utilized to detect heat-responsive QTLs using 205 wheat varieties with a late sowing method identifying a total of 69 potential QTLs across ten different traits, including grain-filling duration and grain-filling rate [144]. Approximately 300 QTL/MTAs for different agronomic and physiological traits have been reported in wheat [145].

Heat tolerance results from a combination of different genes that regulate adaptive characteristics such as enhanced photosynthetic activity, cool canopy, stomatal conductivity, and improved pubescence, as most of these characteristics correlate with improved grain yield under heat stress conditions [9]. Several genomic regions have been reported in wheat utilizing interval mapping (IM) and GWAS for heat-stress-tolerance-related traits such as days to heading, thousand-grain weight, yield, grain-filling duration [146], canopy temperature depression [147], stay-green- and senescence-associated traits [148], and chlorophyll-content-related traits [149]. The QTLs related to heat response in wheat are shown in Table 2.

Multi-omics studies provide lots of potential candidate genes responsible for heat tolerance. Genome-wide analysis has proved to be a valuable method for identifying heat-stress-responsive genes due to the complexity of the underlying heat tolerance mechanisms [156]. Recent advances in wheat gene transformation technology and transgenic studies have accelerated the evolution of functional analysis of heat-responsive genes in wheat [143]. The functions of some genes have been characterized by the overexpression of genes involved in sensing and responding to heat stress in wheat [157].

According to the transcriptome analysis, the overexpression of heat-induced spliced form of wheat *TabZIP60* (TabZIP60s) was found to improve heat tolerance in *Arabidopsis*. As a transcription factor, TabZIP60s regulates expression patterns of 1104 genes in response to heat stress [158]. In addition, constitutive expression of *TaPEPKR2* in wheat resulted in enhanced tolerance to both heat and dehydration stresses [159]. Overexpressing wheat NAC transcription factor *TaNAC2L* in *Arabidopsis* led to an increased survival rate of seedlings under heat stress conditions [160].

### 5.2. Identification and Exploration of Germplasm Tolerant to Heat Stress

The negative effect of heat stress on wheat production is exacerbated by greater genetic uniformity resulting from the narrowing of the varieties grown in developed countries [161]. This warrants increased efforts to explore new genetic resources and useful traits to counteract the effect of heat stress on wheat productivity. With an estimated 0.8 million wheat genetic resources available in collections worldwide, there is ample opportunity to tap into new sources of abiotic stress tolerance [162].

As a significant resource for genomic studies, landraces are the easiest to breed with resynthesized hexaploid wheat [163], which contains larger genomic variance and genetic resources adaptable to harsh environmental conditions [164]. Several wheat genotypes cultivated globally and well adapted to abiotic stresses have been bred from landraces. For example, bread wheat variety Aragon 03, selected from an indigenous landrace population in Spain, remains widely cultivated due to its ability to tolerate abiotic stresses [165]. Wild emmer wheat (*T. turgidum* ssp. dicoccoides) is considered a viable genetic resource due to its direct lineage to domesticated durum wheat (*T. durum*) and the A and B genomes of bread wheat (*T. aestivum*), which also encompasses significant agronomic, physiological, and yield-related characteristics that are linked to the ability to withstand heat stress [166]. The genetic introgression of wild emmer wheat for wheat improvement has been demonstrated to be feasible through the two lineages it possesses.

Yield and yield-contributing phenological and physiological characteristics differ among wheat genotypes. Hays et al. identified significant differences in genotype performance, with heat-susceptible genotype Karl 92 showing a decrease in kernel weight of up to 28.3% compared to a non-significant response in Halberd [167]. Significant differences were also found in the genotype response to heat stress conditions related to leaf senescence and leaf chlorophyll content [168]. Chlorophyll fluorescence and membrane thermostability are indicators of high-temperature stress tolerance in hard white Australian wheat Ventor [169], as they have strong genetic correlations with grain yield. It is confirmed that improved terminal heat tolerance is linked to the stay-green trait in bread wheat genotypes [170]. Wheat genotypes expressing HSPs can withstand terminal heat stress better than those not expressing heat-shock proteins [171]. Nowadays, no consistent selection criterion has been established to evaluate diverse genetic materials for tolerance to heat stress. Selection criteria and screening methods for identifying heat-tolerant wheat genotypes are generally approached based on reliable yield efficiency and relative performance of yield-contributing traits [172]. In this regard, researchers suggest some indirect selection criteria for developing heat tolerance in wheat such as photosynthetic rate, grain weight, and membrane stability (Table 3).

The occurrence of heat stress is always accompanied by drought stress, which exerts detrimental effects on various physiological, growth, and developmental processes in wheat, such as photosynthesis, respiration, and grain-filling duration [180]. Drought stress occurs due to a decrease in soil water content, leading to reduced water availability, triggering turgor loss and inhibiting photosynthesis and long-distance transport. In contrast, heat stress results from elevated temperatures and is intensified by an increase in solar radiation. Most studies have tended to assume that there is a common response to drought or heat stress and do not consider stress tolerance as the ability of plants to respond to abnormal environmental conditions [181]. In order to distinguish between traits associated with drought tolerance and heat tolerance, we present some of the main physiological traits that can currently be applied in wheat breeding to improve heat and drought adaptation (Table 4).

The regulation of stomata is the primary mechanism of the avoidance response to both drought and heat, where decreases and increases in transpiration enable water conservation and foliar cooling, respectively. Drought tolerance response is represented by the activation of ROS scavenging pathways, increased biosynthesis of compatible solutes, and the accumulation of protective molecules [182,183,184]. Generally, soluble metabolites accumulate in response to both drought and heat, but the profiles of metabolites are specific to drought, heat, and the combination of these stresses and depend on the plant species [185,186,187]. Under drought stress, to decrease the water potential in cells and prevent the loss of water, osmoprotectants such as sugar, amino acids, and ammonium compounds accumulate [188,189]. In addition, malondialdehyde exhibited a positive direct effect on grain yield under drought stress but a negative direct effect under other stress conditions [190].
plants-14-00148-t004_Table 4Table 4Different responses under drought and heat stress conditions.TraitsHeat StressDrought StressReferencesStomatal responseOpenClose[191]TranspirationIncreaseDecrease[191]PhotosynthesisHigh CO_2_Net photosynthesis reductionLow CO_2_[192,193]
Impaired carbon fixation, photorespiration occurrenceROS productionElectron leakage produces ROSReduction in CO_2_ availability results in the accumulation of ROS[194,195]Protective proteinsHSPsLate embryogenesis abundant (LEA) proteins[196]

### 5.3. Breeding Wheat Cultivars with High Heat Stress Tolerance

Breeding cultivars with high heat stress tolerance is the most efficient and fundamental approach for coping with global warming in crop production. With the rapid development of molecular biology including omics and gene-editing technologies, new breeding methods, such as marker-assisted selection and genomic selection, have been used in developing wheat cultivars alongside conventional breeding, casting a hopeful light for successful breeding.

#### 5.3.1. Hybrid Breeding

Wild relatives of commercial wheat cultivars provide additional sources of variation for breeding efforts [197]. Field studies have shown that lines derived from crosses with synthetics can not only display outstanding yield but also express a range of physiological traits under heat stress [198]. However, the widespread adoption of hybrids has been limited due to the cost of seed. Although several systems for generating male sterile wheat are available, the strong inbreeding structure of the wheat flowers has made most of the techniques difficult and unreliable [199]. Until efficient and low-cost methods for the large-scale production of hybrid seed are available, hybrids are likely to be restricted to high-yielding environments where the costs can be justified.

#### 5.3.2. Marker-Assisted Selection

High-throughput marker-assisted selection (MAS) breeding can fast-track plant breeding with high productivity [200]. Marker-helped backcrossing (MABC), marker-helped intermittent choice (MARS), GWAS, and genomic determination are all examples of MAS-based methods [201]. GWAS and other QTL mapping methods related to heat stress tolerance characteristics can help develop wheat cultivars that are ideal for high-temperature climates [153]. In general, QTLs elucidated in environments, known as constitutive QTLs, could be used to develop heat-tolerant landraces. On the other hand, QTLs detected alone in purposefully designed environments, called adaptive QTLs, could be utilized for specific heat-stressed areas [202]. Mapping/identification of QTLs is a recently developed tool for combining genetic information with phenotypic measures to unpin the genetic basis of stress tolerance in crop plants, including wheat [203]. Mapping QTLs linked with heat stress tolerance, using marker-assisted selection, has identified mechanisms of heat tolerance in wheat grown in high-temperature environments [153]. For instance, QTLs for heat stress tolerance have been identified for grain yield and yield-related traits, including grain weight/number per spike, 1000-grain weight, and grain-filling rate and duration [204].

#### 5.3.3. Genomic Selection

Genomic selection (GS) is a modern approach to plant breeding that involves the use of DNA markers to predict the breeding values of individuals for different traits. GS is effective in promoting novel cultivars in many crops [205]. This technique has gained considerable attention in wheat breeding because of the complexity of the wheat genome and the need to develop improved varieties that can withstand various biotic and abiotic stresses. The development of high-density SNP arrays has facilitated the application of genomic selection in wheat breeding. The foremost gain of GS compared to MAS is that minor-effect alleles are also detected and utilized in the marker selection process [206]. GS is a helpful strategy for preparing novel breeding and advancing ground-breaking genomic evaluation marker-based models. The accessibility of high-throughput, cost-effective genome-wide, and scalable molecular markers suitable for a large population, with or without the reference genome, is essential for the successful application of genomic selection in crops [207]. GS can be used as a pre-breeding tool for detecting genomic resources with an advantageous variation for compound traits by predicting the breeding values of a specific population within the breeding population. It also offers new prospects to upsurge genetic gain for complex traits [208]. GS has been applied in successful breeding programs in wheat [209]. In the past decade, numerous studies have investigated the potential of genomic selection in wheat, particularly for yield-related traits, disease resistance, and abiotic stress tolerance [210,211]. These studies have shown promising results, indicating that genomic selection could effectively predict wheat improvement, enhance breeding efficiency, and accelerate the development of improved wheat varieties. The utilization of GS for the breeding of wheat to adapt to heat-stressful environments is also employed in several breeding institutes, companies, and universities [210].

#### 5.3.4. Genetic Engineering and Gene Editing

Genetic divergence is a vital technique for breeding new cultivars based on genetic resources [212]. Additionally, various biotechnological approaches like gene editing alongside the latest advanced omics tools can aid in making heat-stress-tolerant cultivating varieties [213].

Transgenesis involves transferring superior genes to candidate wheat genotypes, which avoids linkage drag involving the co-transfer of unwanted adjacent gene segments or exploiting genes not accessible in hybridization-based breeding [214]. In wheat, transgenic methods and genetic modification can increase terminal heat tolerance after inserting genes of interest in the candidate genotype [215]. The protein synthesis elongation factor (EF-Tu) present in chloroplasts correlates with wheat heat tolerance, improving heat stress tolerance for a longer period [216]. The constitutive expression of *EF-Tu* in transgenic wheat protects leaf proteins against thermal degradation, decreases thylakoid membrane disruption, resists pathogenic microbe infection, and enhances photosynthetic capacity [216]. It is reported that transgenic wheat overexpressing the maize *EFTu1* gene has increased heat tolerance [217]. Wheat is a complex crop with a large genome, and many genes contribute to heat tolerance. As a reverse genetic strategy for targeting heat tolerance gene activity, genome-editing techniques such as TALENs, ZFNs, and CRISPR can also be used in addition to MAS. Compared to other genome-editing techniques, CRISPR-Cas9 has developed a powerful method for precise genome editing to study the pathways associated with heat stress and to increase thermo-tolerance in cropping systems [218]. It can modify target genes by insertion, deletion, and knock-in/knock-out alterations, enhancing agricultural plants’ capacity to scavenge ROS. Genome editing methods have unlocked new opportunities to initiate targeted editing of crop genomes involved in heat stress tolerance [219,220].

## 6. Conclusions and Prospects

Heat stress caused by global warming has posed a huge threat to the production of crops, in particular wheat, which is relatively sensitive to high temperatures. Exposed to heat stress, wheat plants suffer from various damages, including disrupted cell structure, impeded metabolisms, and oxidative stress. This results in shortened growth duration, early senescence, and reduced biomass production, consequently resulting in decreased grain yield and deteriorated quality. On the other hand, there is a large genotypic variation in the response or tolerance to high temperature, which provides the possibility and genetic resources for developing wheat cultivars with high heat stress tolerance. Morphological, physiological, and molecular changes are involved in adaptation to heat stress, and the relevant genes or QTLs have been identified and used in wheat breeding.

Breeding is dependent on accessing and utilizing genetic diversity and new techniques in genotyping and phenotyping. AI technologies, including machine learning, deep learning, and high-throughput phenotyping, enable the fast and accurate analysis of large genetic and environmental datasets, improving the breeding process [221]. Breeding and the selection of traits, such as grain weight, grain number, stay-green trait, osmolyte accumulation, and antioxidant enzymes, can be useful for improving wheat performance under terminal heat stress. Genetic engineering that identifies heat-responsive genes/transcription factors and QTLs linked to terminal heat stress tolerance may be another viable option for improving wheat performance under HS. New high-throughput phenomics techniques reduce the tedium of measuring difficult-to-measure module characters associated with heat tolerances. When used with systems biology techniques, new omics-based applications could enormously improve conventional breeding to mitigate the impact of heat-stress-induced yield reduction in wheat and modernize the future of sustainable agriculture. Researchers around the world are doing their best to develop heat-tolerant wheat genotypes with higher yields under changing climatic conditions to ensure food security for generations to come.

## Figures and Tables

**Figure 1 plants-14-00148-f001:**
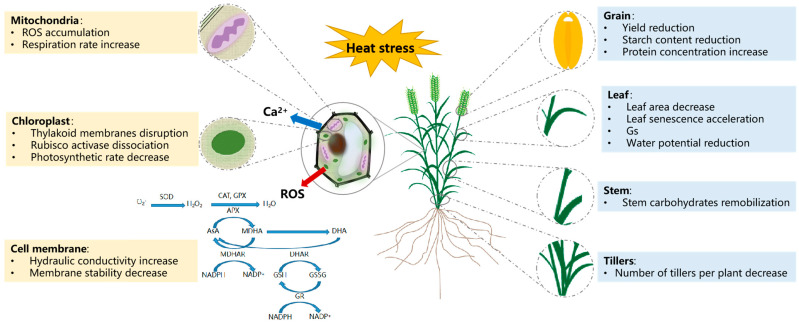
Effects of heat stress on wheat and its response.

**Table 1 plants-14-00148-t001:** The functions of different heat shock proteins (HSPs) in plant heat tolerance.

HSPs	Characteristics		References
Small HSP	Including class I and class II, restores soluble cell fraction, prevents heat aggregation, and protects translation factors		[109,110,111]
HSP40	Stronger response to biotic stress	As a molecular chaperone for HSP70, assists in the folding, translocation, and degradation of proteins	[112,113]
HSP60		Along with HSP10, supports mitochondrial and chloroplast protein folding, assembly, and transportation.	[114,115]
HSP70	Prevents protein aggregation, dissolves aggregated proteins, and stimulates the refolding of misfolded proteins.		[112,116]
HSP90	Regulates protein metabolism, ensures protein stability, and responds to heat-stress-related signal transduction		[117,118]
HSP100	Prevents protein aggregation, aids in clearing or repairing misfolded proteins, and enhances heat tolerance		[119,120]

**Table 2 plants-14-00148-t002:** Summary of important QTLs associated with heat-tolerance-related traits in wheat.

Trait	QTL	Marker	Mapping Approach (QTL-IM/GWAS)	Chromosome	Population Type (Size)	References
Days to heading	Qdh.ccshau-2A	xgwm512-xgwm448	IM	2A	RIL (80)	[142]
	S5B_586352552	-	GWAS	5B	-(125)	[150]
	S7A_3066534	-	GWAS	7A	-(125)	[150]
	S7D_6002850	-	GWAS	7D	-(125)	[150]
	QDTH-6D.1	GENE-4153_101-D_GCE8AKX01CWZ8Z_144	IM	6D	RIL (276)	[151]
Plant height	S2A_748204192	-	GWAS	2A	-(125)	[150]
	Qph.ccshau-2A	xgwm512-xgwm448	IM	2A	RIL (80)	[142]
	QPH-5B.1	Ku_c10415_662-TA003058-0693	IM	5B	RIL (276)	[151]
Grain number per spike	S2A_1050029	-	GWAS	2A	-(125)	[150]
	S5D_503657305	-	GWAS	5D	-(125)	[150]
	QGNP-HS-R1	AX-95652063-AX-95660318	IM	1A	FST (277)	[152]
Grain-filling duration	QHthsigfd.bhu-2B	Xgwm935-Xgwm1273	IM	2BL	RIL (148)	[153]
	QHgfd.iiwbr-5A	X1079678|F|0	IM	5A	F_2_(140)	[154]
Spikes number per plant	S2D_72213516	-	GWAS	2D	-(125)	[150]
	-	wPt-2883	GWAS	7B	-(188)	[155]
Thousand-grain weight	S6B_680699350	-	GWAS	6B	-(125)	[150]
	QTGW-2A.1	2264948|F|0-9:T>A-9:T>A-Kukri_c22235_1547	IM	2A	RIL (276)	[151]
	QHthsitgw.bhu-7B	Xgwm1025–Xgwm745	IM	7BL	RIL (148)	[152]
Grain yield	S6A_340738287	-	GWAS	6A	-(125)	[150]
	QYLD6D.1	2265648|F|0-60:A>G-60:A>G-RAC875_c57371_238	IM	6D	RIL (276)	[151]
	QlsYLD.bhu-7B	Xgwm1025–Xgwm745	IM	7BL	RIL (148)	[152]
Grain yield per plant	QGYP-HS-R1	AX-111105973-AX-94402739	IM	1A	FST (277)	[152]
Fv/Fm	QHst.cph-3B.2	Xgwm389	IM	3B	F_2_(140)	[154]

**Table 3 plants-14-00148-t003:** Potential characteristics for selecting wheat for tolerance to heat stress.

	Characteristics	References
Morphological and growth	Leaf senescence	[173]
	Accumulation of biomass	[10]
	Leaf area expansion	[26,28]
	Stem reserves remobilization	[174]
	Biomass production	[10]
Photosynthetic apparatus	Photosynthesis rate	[25]
	Stomatal conductance	[63]
	Leaf chlorophyll levels	[17]
	Chlorophyll fluorescence	[175]
	Spike photosynthesis	[9]
Physiology and biochemistry	Water potential	[21]
	Respiratory rate	[176]
	Leaf canopy temperature	[177]
	Membrane thermostability	[174,177]
	Antioxidant activity	[11]
Growth stages	Days to heading	[153]
	Days to maturity	[177]
	Stay green duration	[178]
	Grain-filling duration	[153]
Yield	Number of tillers per plant	[177]
	Number of grain yield per spike	[177]
	Number of spikes per plant	[177]
	1000-grain weight	[153]
	Harvest index	[179]
	Setting rate	[51]

## Data Availability

No new data were produced in this research.

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
