# Peer review of "The Genetics and Breeding of Heat Stress Tolerance in Wheat: Advances and Prospects"

_plants, 2025, doi:10.3390/plants14020148_

Round 1

Reviewer 1 Report

Comments and Suggestions for Authors

The review article "The Genetics and Breeding of Heat Stress Tolerance in Wheat: Advances and Prospective” provides a comprehensive overview of the various aspects associated with heat stress tolerance in wheat. The review touches on various aspects related to heat stress, including the impacts of heat stress on growth and yield, the responses of wheat plant to heat stress, the molecular mechanisms of heat stress tolerance in wheat, and breeding for heat tolerance in wheat.

Understanding of physiological and biochemical mechanisms related to heat stress tolerance is crucial for identifying genetic basis of heat stress and determining ways to cope with this stress. This is a vast area of research and involves research studying the effect of heat on grain yield and quality, antioxidant defense systems, photosynthesis and chlorophyll content, plant canopy, hormones effected by heat, heat shock proteins among other.

The knowledge of genes or QTLs regulating heat stress tolerance, identification of germplasm resistant to heat stress can be utilized in wheat breeding to develop varieties that are resistant to heat stress. The novel breeding approaches like genomic selection approach and gene editing are also becoming important and precise tools to improve new genetics. The use of genomic selection and gene-editing for heat stress tolerance is also discussed in this review.

Overall, the review captured these important areas of research associated with heat stress tolerance in wheat. The review is well written and easy to follow.

Author Response

 Comment

The review article "The Genetics and Breeding of Heat Stress Tolerance in Wheat: Advances and Prospective” provides a comprehensive overview of the various aspects associated with heat stress tolerance in wheat. The review touches on various aspects related to heat stress, including the impacts of heat stress on growth and yield, the responses of wheat plant to heat stress, the molecular mechanisms of heat stress tolerance in wheat, and breeding for heat tolerance in wheat. Understanding of physiological and biochemical mechanisms related to heat stress tolerance is crucial for identifying genetic basis of heat stress and determining ways to cope with this stress. This is a vast area of research and involves research studying the effect of heat on grain yield and quality, antioxidant defense systems, photosynthesis and chlorophyll content, plant canopy, hormones effected by heat, heat shock proteins among other. The knowledge of genes or QTLs regulating heat stress tolerance, identification of germplasm resistant to heat stress can be utilized in wheat breeding to develop varieties that are resistant to heat stress. The novel breeding approaches like genomic selection approach and gene editing are also becoming important and precise tools to improve new genetics. The use of genomic selection and gene-editing for heat stress tolerance is also discussed in this review. Overall, the review captured these important areas of research associated with heat.

Authors' anwser: Thank you for your positive evaluation on this review. Hopefully it will be accepted for publication, providing the valuable information to the readers soon.

Reviewer 2 Report

Comments and Suggestions for Authors

The authors referenced approximately 200 papers in their review, "The Genetics and Breeding of Heat Stress Tolerance in Wheat: Advances and Prospective." Heat stress poses a significant challenge to wheat production globally. This review comprehensively addresses the impacts of heat stress on plant growth and grain yield, wheat's physiological and genetic responses to heat stress, key QTLs and genes associated with heat tolerance, and breeding strategies to mitigate these challenges.

The review is well-written, thoroughly formatted, and concise. However, there are two minor comments to address:

Line 29: Replace "yield" with "grain yield."

Section 5.2: Add a discussion on how to distinguish between traits associated with drought tolerance and heat stress tolerance. Since these traits may overlap, include suggestions for improved phenotyping methods to better assess germplasm for these specific tolerances.

Author Response

Comment 1: The authors referenced approximately 200 papers in their review, "The Genetics and Breeding of Heat Stress Tolerance in Wheat: Advances and Prospective." Heat stress poses a significant challenge to wheat production globally. This review comprehensively addresses the impacts of heat stress on plant growth and grain yield, wheat's physiological and genetic responses to heat stress, key QTLs and genes associated with heat tolerance, and breeding strategies to mitigate these challenges.   The review is well-written, thoroughly formatted, and concise.

Authors' answer: Thank you for your positive evaluation on this review.

Comment 2: However, there are two minor comments to address:       Line 29: Replace "yield" with "grain yield."

Anthors' answer: Thank you for your suggestion. We changed yield by grain yield.

Comment 3: Section 5.2: Add a discussion on how to distinguish between traits associated with drought tolerance and heat stress tolerance. Since these traits may overlap, include suggestions for improved phenotyping methods to better assess germplasm for these specific tolerances.

Authors' answer: A valuable suggestion! We added a discussion according to your suggestion in the revised version.

Reviewer 3 Report

Comments and Suggestions for Authors

This paper is a good contribution to the field,\. In line 162 suggest substituting

degrade" for 'dissolve".

Author Response

Comment1 : This paper is a good contribution to the field,\. In line 162 suggest substituting degrade" for 'dissolve".

Authors' answer: Thank you for your positive evaluation on this review, and we changed the word according to your suggestion.